# Encapsulating TGF-β1 Inhibitory Peptides P17 and P144 as a Promising Strategy to Facilitate Their Dissolution and to Improve Their Functionalization

**DOI:** 10.3390/pharmaceutics12050421

**Published:** 2020-05-02

**Authors:** Nemany A. N. Hanafy, Isabel Fabregat, Stefano Leporatti, Maged El Kemary

**Affiliations:** 1Nanomedicine Department, Institute of Nanoscience and Nanotechnology, Kafrelsheikh University, Kafrelsheikh 33516, Egypt; elkemary@yahoo.com; 2Bellvitge Biomedical Research Institute (IDIBELL), University of Barcelona (UB) and CIBEREHD, Gran Via de l’Hospitalet, 199, Hospitalet de Llobregat, 08908 Barcelona, Spain; ifabregat@idibell.cat; 3CNR NANOTEC-Istituto di Nanotecnologia, Via Monteroni, 73100 Lecce, Italy

**Keywords:** transforming growth factors, proliferation, polymer hybrid protein

## Abstract

Transforming growth factor-beta (TGFβ1) is considered as a master regulator for many intracellular signaling pathways, including proliferation, differentiation and death, both in health and disease. It further represents an oncogenic factor in advanced tumors allowing cancer cells to be more invasive and prone to move into the metastatic process. This finding has received great attention for discovering new therapeutic molecules against the TGFβ1 pathway. Among many TGFβ1 inhibitors, peptides (P17 and P144) were designed to block the TGFβ1 pathway. However, their therapeutic applications have limited use, due to lack of selection for their targets and their possible recognition by the immune system and further due to their potential cytotoxicity on healthy cells. Besides that, P144 is a highly hydrophobic molecule with less dissolution even in organic solution. Here, we aimed to overcome the dissolution of P144, as well as design nano-delivery strategies to protect normal cells, to increase cellular penetration and to raise the targeted therapy of both P17 and P144. Peptides were encapsulated in moieties of polymer hybrid protein. Their assembly was investigated by TEM, microplate spectrum analysis and fluorescence microscopy. SMAD phosphorylation was analyzed by Western blot as a hallmark of their biological efficiency. The results showed that the encapsulation of P17 and P144 might improve their potential therapeutic applications.

## 1. Introduction 

Transforming growth factor (TGFβ) is a secreted cytokine, having the ability to regulate and control cell proliferation, migration, differentiation and cytoskeleton morphology [1,2,3]. Apart from this fact, the role of TGFβ in controlling inflammation, wound healing and tissue repair received a lot of interest [4]. However, its function as a tumor promoter at the end stage of cancer development resulted in an impact issue, since it supports cancer growth, activates tumor angiogenesis and inhibits immune responses [5,6,7]. Among many molecules that were used to inhibit TGFβ signaling pathway, TGFβ inhibitory peptides have obtained great interest due to their efficient role in blocking of TGFβ signaling pathways [8]. Peptide P144, TSLDASIIWAMMQN, is a very hydrophobic peptide obtained from the membrane-proximal ligand-binding domain of b-glycan [9]. This peptide is designed to block TGFβR III extracellular domains preventing cellular interaction between TGF ligand and its receptors [10]. Another soluble peptide is called P17, (KRIWFIPRSSWYERA) [11]. It was produced from a phage library [12]. P17 can block TGF-β1, TGFβ2 and TGFβ3 with relative affinity binding reached 100%, 80% and 30% respectively [13]. The active inhibitory effect of both peptides was characterized in vivo and in vitro for several models of fibrosis and scleroderma [14]. Results have proven the potential therapeutic value for both peptides to block the TGFβ pathway and to prevent the accumulation of collagen fibers [15]. However, there is an urgent needing strategy to improve their dissolution, prevent their aggregation and facilitate their delivery into animal models. P144 was used previously either after it is suspended inside dimethyl sulfoxide (DMSO)-saline [16] or, after its integration into the composition of the lipogel in the presence of 5% DMSO [17]. Both strategies were restricted due to the presence of DMSO [18]. Additionally, both peptides can be distributed into the whole body, with no specific delivery into a certain region. Leading to increase their accumulation inside healthy tissues. Additionally, due to their amino acid structure, they can be recognized in the bloodstream and then can be engulfed by the immune system or can be degraded inside the stomach by a biological enzyme [19]. In the current study, the sonicated P144 and suspended P17 were internalized into the bovine serum albumin matrix through amino–carboxyl interaction. Such attachment is characterized by strong interaction between peptide and protein due to the presence of carboxyl, amino-groups and hydrogen intermolecular interactions.

Additionally, the surface of the protein–peptide complex was further functionalized by folic-acid-attached carboxymethyl cellulose (CMC; Appendix A). Folic acid is used as a ligand and can bind folate receptors. Additionally, CMC has mucoadhesive properties and allows protein–peptide formulation to adhere and penetrate mucus layers. This strategy provides a novel and concrete reason to strengthen the potential application of peptides as a targeted delivery. The efficiency of encapsulated peptides (P144 and P17) and the pure peptides (with no addition of DMSO or integration into lipogel) were studied by using two different hepatocellular carcinomas (HCC) cell lines: hepatitis B-positive SNU449 cells, [20], that were characterized by a mesenchymal phenotype expression [21] and the human epithelial HCC Hep3B cells [22] with different genetic characterization. 

## 2. Materials and Methods 

### 2.1. Chemicals

The suppliers of the chemicals were as follows. Carboxy methylcellulose (CMC) was purchased from Fluka, Sigma-Aldrich (St. Louis, MO, USA), phosphate-buffered saline (PBS) tablets of pH 7.3 were purchased from Oxoid Limited Basingstoke (Hampshire, England); ethanol from Baker Analyzed, Fisher Scientific, (Landsmeer, The Netherlands); bovine serum albumin (BSA), folic acid (FA), crystal violet, 40,6-diamidino-2-phenylindole dihydrochloride (DAPI), paraformaldehyde, N-ethyl-N0- (3 dimethylaminopropyl)carbodiimide hydrochloride (EDAC), N-hydroxy succinimide (NHS), dimethyl sulfoxide (DMSO) from Sigma-Aldrich (St. Louis, MO, USA).

### 2.2. Fabrication of Protein–Peptide Mucoadhesive Carriers 

A 1 mg amount of sonicated P144 and suspended P17 was mixed into BSA solution (50 µg/50 mL). Then, the mixture will be grafted by carboxymethyl cellulose inside sterilized glass vials for 30 min under rotation by a magnetic stirrer in the presence of EDAC/NHS. The complex was further coated by 5 mL of 50 µg/50 mL protamine and the stirrer was continued into addition for 10 min. Afterward, the mixture was furthermore covered by folic-acid-attached carboxymethyl cellulose and the rotation was completed in an additional 15 min. Un-reacted materials were removed by using a dialysis bag against distilled water for 48 h. The water was changed several times. Then, the final product was lyophilized at a freeze-drying machine while the material was kept at −20 °C until use.

### 2.3. Characterization of the Protein-Peptides Mucoadhesive Formulation

#### 2.3.1. Transmission Electron Microscopy (TEM)

A 10 µL volume of nanoparticle suspension was deposited on the copper grid and air-dried before measurement. Copper grids sputtered with carbon films were used to support the sample. High-resolution TEM images of nanoparticles were analyzed by a Hitachi HT 7700 operating (Hitachi, Japan) at 100 kV, coupled with a GATAN camera ORIUS SC600 (Gatan, Inc., Pleasanton, CA, USA)) with a resolution of 7 megapixels. The GATAN camera is controlled by Digital Micrograph (Gatan, Inc., Pleasanton, CA, USA) [23]

#### 2.3.2. UV–Vis Spectroscopy 

The absorbance of folic acid and rhodamine was measured by using a multiplate reader at the option of UV Absorbance Spectrophotometer in multiplate wells. A 500 µL mL volume of fabricated nanoparticles was scanned at a range of 300–600 nm [24].

#### 2.3.3. Zeta Potential 

The electrophoretic mobility of samples was determined by photon correlation spectroscopy by using a Zeta Nano Sizer (Malvern Instruments, Malvern, UK). All measurements were performed at 25 °C. Five following measurements were taken for analysis; each was run five times with a delay of 5 s after each measurement [25].

#### 2.3.4. ImageJ Analysis and Calculation of the Surface Roughness

The measurements for the light microscopic images (400 µm) were carried out in pixels. Thus, setting up a scale bar for calibration for the calculation of the actual area was not required. Since, in the “Image” option of ImageJ, under Adjust, “size” was selected. Then, the pixel of width and length was adjusted and it is followed by using the “Surface Plot” option in “Analyze”. This led to the area measurement of the complete image in pixels.

The measured peaks upon the material surface expressed the status of material that was either roughness or granules, while no peaks were measured in the case of soft materials or those that did not contain any granulated materials [26]. 

### 2.4. Cell Culture

Hep3B and SNU449 cells were obtained from the European Collection of Cell Cultures (ECACC). Cell lines were never used in the laboratory for longer than 4 months after receipt or resuscitation. All cells were maintained in DMEM media (Lonza, Basel, Switzerland) supplemented with 10% FBS (Sera Laboratories International Ltd., West Sussex, UK), penicillin (100 U/mL), streptomycin (100 μg/mL), amphotericin (2.5 μg/mL) and L-glutamine (2 mM). They were maintained in a humidified atmosphere at 37 °C and 5% CO_2_. Cells were observed under an Olympus IX-70 microscope.

#### 2.4.1. Cellular Uptake 

Ten thousand human liver cancer cell lines were seeded upon the surface of a sterilized cover slip that was laid in the bottom of 6 multi-well microplates. After 24 h from their growth, encapsulated peptides and non-encapsulated (50 µg/mL) were added to each well and incubated in a humidified atmosphere of 37 °C, 5% CO_2_. SNU449 and Hep3B cell lines were fixed by 4% paraformaldehyde then washed by PBS (phosphate-buffered saline) at pH 7.2. Cells were then stained by DAPI (nuclear stain) for 30 min and washed twice by PBS,7.2. Cellular uptake was analyzed after 24 h by red (TRITC), green (FITC) and blue (DAPI) channels of fluorescence microscopy. After that, images were captured by fluorescence microscopy with a digital camera [27,28].

#### 2.4.2. Proliferation Assay 

To evaluate cell proliferation, crystal violet assay was performed at 3, 6, 12, 24, 48 and 72 h [29]. Briefly, 7 × 10^3^ cells/well were seeded in 96-well flat-bottom plates and allowed to grow for 24 h. Encapsulated peptides (P17 and P144) and free capsules were then added at different concentrations (10, 50, 100, 200 µg/mL). After incubation for a certain time, DMEM media was removed and cells were washed using PBS 3 times. Cells were then fixed for 10 min in a solution of buffered formalin (3.7%), and then cells were washed with PBS (pH 7.3) and subsequently stained with a 0.01% crystal violet solution. After removing excess stain, the crystal violet stained cells were dissolved in 1 mL of a 10% sodium dodecyl sulfate solution for 2 h under orbital shaking and the optical density of the extracted dye was read with a spectrophotometer at 590 nm. Optical density measurements give an indication of the relative number of viable cells present at the time of adding dye and this is used to create survival curves. Cell survival at each dose point was expressed as a percentage of the control survival rate.

#### 2.4.3. Western Blot Analysis

Total protein extracts were obtained as described previously [30], separated by SDS/PAGE (12% polyacrylamide gels) and transferred on to PVDF membranes. After blocking with 5% (*w/v*) non-fat milk TBST (Tris-buffered saline solution containing 0.05% (*v/v*) Tween 20), the membranes were incubated overnight with the corresponding antibody in a 0.5% non-fat milk TBST (diluted 1:5000 for β-actin and 1:1000 for all others). After washing and incubating the membrane with an appropriate peroxidase-conjugated antibody (diluted 1:5000) for 1 h at 21 °C, antibody binding was revealed using ECL^®^ (GE-Healthcare). β-actin was used as a loading control.

### 2.5. Ethical Approval

All experiments were conducted in accordance with US National Institutes of Health Guidelines for the Care and Use of Laboratory Animals and cell line experiments Guide for the Care and Use of Laboratory Animals after being approved by the relevant Ethical Committee and authorized by the Italian and German Ministry of Health. This study was also approved by the Research Ethics Committee of Kafrelsheikh University.

## 3. Results

### 3.1. Physical Properties of Peptide P17 and P144

P144 is a very hydrophobic peptide, has a limited dissolution either inside aqueous solution or even in an organic solvent such as DMSO (Figure 1A,B) [10]. This drawback leads to a reduction of its delivery into preclinical studies and minimizes its potential applications. The dissolution of bulk P144 was obtained by using physical sonication for 15 min under high amplitude to prevent microscopic bubbles in solution. After sonication, the bulk of P144 was separated into a colloidal suspension containing small peptide particles (Figure 1C,D). However, these small particles turned into aggregation after their precipitation (Figure 1G). The bulk of P144 is formed by tiny small molecules held together by electrostatic force. These small particles had lost their ionic balance after their separation. For this reason, the separated particles of P144 tend to bind back forming an aggregated state. Contrarily, P17 is very soluble in water, forming a colloidal suspension with good stability [11]. Meanwhile, P17 has a widescreen attachment with three TGFβ isoforms such as TGFβ1, TGFβ2 and TGFβ3. 

To investigate their properties with fluorophore conjugation and to follow their cellular uptake, the anti-TGFβ inhibitory peptides (P17 and P144) were labeled by rhodamine (RG6; Figure 1G) [31]. Rhodamine-labeled peptides showed strong absorbance at 550 nm associated with a peak of pure rhodamine [32]. Their interaction is attributed to the amino group of rhodamine and the carboxyl group of peptides. It is noticed that rhodamine-doped peptides have a strengthened second peak localized at 490 nm. This peak emitted green fluorescence with fluorescence microscopy (Figure 1G,H). In addition, sonicated P144 and dissolved P17 were shown as aggregated peptides even after their cellular uptake (Figure 1H).

### 3.2. Fabrication of Protein–Peptide Mucoadhesive Complex 

Both peptides (P17 and 144) were suspended in PBS at pH 7.2 and then P144 was sonicated by ultrasonicator. The processing time was adjusted for 15 min under amplitude five (intensity). The processing time is further divided into three steps each one has taken 5 min. Under magnetic stirrer, the sonicated P144 and suspended P17 were then integrated into BSA moieties in the presence of carbodiimide and n-hydroxy succinimide [33]. The reaction was continued by completing the stirrer for 15 min and then carboxymethyl cellulose was added and the stirring was further continued into addition for 10 min. Free capsules (with no encapsulated peptides) were labeled by fluorescence isothiocyanate (Figure 5 and Appendix A). Both formulations were then functionalized by adding protamine to observe helix structure [34]. Indeed, protamine can modify the size of particles without any restriction for moieties of the protein–peptide complex. These results showed good turbidity after using protamine due to the presence of arginine (Figure 2A,B) [34]. Peptide protein mucoadhesive nanoparticles exhibit unaggregated and rounded core-shell assembly (Figure 2C–E) and have Z-potential value (−25 mV; Figure 3). This indicates good physical stability of nanosuspensions due to electrostatic repulsion of individual particles. Additionally, folic acid doped carboxymethyl cellulose was added as the last layer to the formulation. This may strengthen the properties of mucoadhesive targeting therapy (Figure 2F,G and Appendix A). Indeed, this strategy was designed by using two layers from CMC, one is integrated into the middle layer and the second is added as the last layer. Therefore, this mechanism allows facilitating cellular adhesion and penetration, leading to increased stability of peptides in aqueous solution and improved cellular uptake for both peptides (P144 and P17; Appendix A). 

### 3.3. Biological Experiments: Cellular Uptake 

The rhodamine-labeled non-encapsulated peptides were accumulated successfully inside the cytoplasm as shown by fluorescent emission of TRITC channel (red spots). On the other hand, the green color was shown by using the FITC channel, indicating that peptides are able to emit green intensity after their doping with R6G (Figure 1G,H). Similarly, encapsulated peptides that were labeled by rhodamine showed good intensity for both TRITC and FITC channels after their cellular internalization as well (Figure 4A−P). Indeed, the labeled peptides were accumulated around the perinuclear region of both Hep3B and SNU449 cell lines. Such emissions are mostly related to the presence of tryptophan, tyrosine and phenylalanine. While in the presence of rhodamine, the fluorescent signal is strengthened and intensity is increased. It is reported that tryptophan is highly sensitive to hydrogen bonding and non-covalent interactions, displaying a red, green and blue shift [35]. Additionally, its indole group is considered the dominant source of UV absorbance at ∼280 nm and emission at ∼350 nm [36].

The results indicate that there is a good distribution of adsorbed peptides inside cytoplasm after their encapsulation compared to those used with no encapsulation. Similarly, free capsules were localized inside cytoplasm after their cellular adsorption (Figure 5). On the contrary, the non-encapsulated peptides accumulated in an aggregated state even after their cellular uptake (Figure 1H (merged)). This perhaps explains that the separation of P144 bulk into small particles by using physical sonication is not really enough to keep these particles separately because these particles tend to bind back after their accumulation.

### 3.4. Determination of Nuclear Morphology by Using DAPI Staining

DAPI (4′, 6-diamidino-2-phenylindole) is a specific fluorophore for a nucleic acid because it is able to bind preferentially to adenine thymine base pair and also to phosphate groups of DNA [37,38,39,40]. This mechanism allows researchers to discover the alteration associated with nuclear morphology. In the current study, DAPI staining indicates that there are no changes in nuclear morphology for cells SNU449 after their exposure to 50 µg/mL free capsules (empty vehicles) with no encapsulated peptides for 24 h (Figure 6). However, there are clear alterations for nuclear morphology in both cell lines Hep3B and SNU449 after their exposure to encapsulated P17 and P144 for 24 h. Results showed that there was an increasing number of condensed nuclei, nuclear fragmentation and hypotrophy [41,42,43]. Additionally, their morphology suffered from phenotype alteration (Figure 7, Part 1 and Part 2).

On the contrary, the Hep3B and SNU449 exposed to non-encapsulated peptides exhibited no more changes in their morphology. However, a slightly condensed nucleation was shown (Figure 8, Part 1 and Part 2). Additionally, quantitative analysis of nuclear morphology indicates that Hep3B and SNU449 cells that were exposed to encapsulated peptides (P17 and P144) exhibited significantly nuclear alterations compared to non-encapsulated peptides (Figure 9).

### 3.5. Antiproliferative Effect of Encapsulated Peptides 

Crystal violet cell proliferation assay is a colorimetric method based on the use of crystal violet as a basic dye with avidity to nuclear structures. After binding and solubilization of the crystal violet, optical density measurements of extracted dye provide a measure of the relative number of viable cells. This test has the advantage that it is less time consuming, easier to perform and more objective [29].

Encapsulated peptides (P17 and P144) have an antiproliferative influence on human liver cancer cell lines (SNU449 and Hep3B) as shown by proliferation assay. Both encapsulated peptides (P17 and P144) inhibited the proliferation of human liver cancer cell lines in vitro in a dose-dependent manner. This inhibition was substantial in both cell lines after treatment, with significant differences from 10 μg/mL to 200 μg/mL in treated cells (Figure 10). Similar to our findings, it is reported that P17 and P144 induced antiproliferation effect against human lung cancer cell lines [44] and glioblastoma cell lines [45,46], respectively. On the other hand, Hep3B and SNU449 cell lines exhibited significant proliferation after their exposure to free capsules for 72 h. 

### 3.6. Effects of P144 and P17 on SMAD2 Phosphorylation 

SMADs are a group of encoded homologs mammalian protein identified by Small (Sma) and Mothers against dpp (Mad) genes that act as intracellular transcriptional factors [47]. SMAD2 and SMAD3 can be phosphorylated by the TGFβRI after the direct binding of TGFβ ligand to its specific receptors (TGFβRII). This enables them to form heteromeric complexes with SMAD4. This complex is translocated into the nucleus to regulate target genes [48]. Indeed, we next studied the response to TGFβ in terms of SMAD2 phosphorylation in both SNU449 and Hep3B cells. SNU449, but not Hep3B, presented basal SMAD2 phosphorylation. Both cell lines responded to external TGFβ significantly increasing phosphor-SMAD2 levels (Figure 11). Additionally, in the absence of external TGFβ1, SNU449 cells presented lower phosphorylation of SMAD2 after their exposure to P17, but not after their exposure to P144. This phosphorylation was not observed in Hep3B cells after their exposure to P17. However, slight phosphorylation was shown after their exposure to P144. In the presence of external TGFβ1, SNU449 cells increased the pSMAD2 signal after their exposure to P17 and P144. However, Hep3B presented lower phosphorylation of SMAD2 after their exposure to P17, but not after their exposure to P144. Interestingly, in the absence of the external TGFβ1, the encapsulated P17 and P144 caused a reduction of SMAD2 phosphorylation in both SNU449 and Hep3B cells. In presence of the external TGF β1, encapsulated P17 clearly attenuated the level of SMAD2 phosphorylation in Hep3B cells and both encapsulated P144 and P17 attenuated SMAD2 phosphorylation in SNU449 cells. These results indicate that the encapsulation of P17 and P144 clearly shows advantages regarding their biological activity when compared with non-encapsulated peptides, particularly in the mesenchymal, more invasive, SNU449 cell line.

## 4. Discussion

TGFβ1 inhibitory peptides have attracted much interest recently because of their ability to block the TGFβ1 signaling pathway [49], due to their possible binding to the extracellular region where the TGFβ1 ligand connects with its cellular receptors [30]. It is reported previously that the bulk of P144 can be separated into small particles by using sonication [30]. However, these small particles tend to be aggregated. This action may attribute their loss of the physicochemical interaction after sonication. Such action leads to the distribution of the electrostatic balance among these particles. For this reason, the small particles tend to be aggregated back after their separation. In the current work, we analyzed the possible separation of these particles by the next interaction with BSA. Hence, these particles are integrated into the moieties of BSA. This coupled interaction drives keeping the small particles of peptides under electrostatic balance and preventing their aggregation. The demonstration of rhodamine-labeled peptides exhibited a specific peak at 550 nm which is attributed to the main peak of pure rhodamine [32]. Increasingly, the second peak was shown at 490 nm, which is similar to the main peaks of fluorescent isocyanate [50]. Perhaps, the central carbon atom of the chromophore is influenced by the ionic stress of the complex leading to a shift of both absorption and fluorescence emission related to the presence of tryptophan [35]. Rhodamine-labeled peptides advanced our knowledge concerning the physicochemical properties of these two peptides. For instance, their aggregation was localized even after their cellular uptake around the cytoplasmic region. This evidence was first detected by using fluorescence microscopy after their precipitation upon the surface of the slide (Figure 1G). In the same way, small particles of both peptides were compacted after their internalization with no equal distribution in the perinuclear region. To evaluate their aggregation, ImageJ analysis was used and it refers to the presence of an aggregation state in non-encapsulated peptides compared to encapsulated peptides (Appendix A).

The protein–peptide assembly was further functionalized by protamine sulfate. This provides a novel strategy in terms of α helix titration resulting in turbidity [34]. Arginine rich protamine is responsible for this turbidity and it perhaps causes modification to the diameter of nanoparticles [51]. Furthermore, there is no published report observing immune-stimulatory side effects related to the use of protamine [51,52]. To increase their adhesion and cellular adsorption, this complex was further functionalized by folic acid conjugated carboxymethyl cellulose. 

Apart from cellular adsorption, the DAPI staining noticeably revealed nuclear morphological changes in both SNU449 and Hep3B human liver cancer cell lines in terms of nuclear fragmentation, nuclear hypotrophy, nuclear condensation and cell structure loss. Accordingly, the present results proved significantly the potential efficient therapy of both encapsulated peptides against the analyzed cell lines. Similar to our results, P144 and P17 treatment resulted in significant inhibition at lung cancer cell lines [44] and the growth of glioblastoma cell lines [46] respectively. When analyzing the effects of encapsulated versus non-encapsulated peptides at the cellular level, we observed that the efficiency of the encapsulated peptides was remarkably higher in the SNU449 cells. We and other authors had previously demonstrated that the SNU449 cells show autocrine stimulation of the TGFβ pathway and respond to TGF-β increasing Smad2/3 phosphorylation [53]. These cells are resistant to the suppressor effects of TGFβ [54] and respond to it undergoing EMT. Here, we show that encapsulated peptides (P17 and P144) induced significant inhibition of basal pSMAD2 in SNU449 cells when compared to non-encapsulated peptides. Additionally, pSMAD2 phosphorylation is considerably attenuated after TGFβ treatment. In summary, the transforming growth factor β (TGF-β) signaling pathway consists of extracellular ligands (including the TGF-β-like group and BMP-like group). The TGF-β-like group generally phosphorylates SMAD2 and SMAD3, whereas the BMP-like group generally induces phosphorylation of SMAD1, SMAD5 and SAMD8 [55]. The activated R-SMADs form hetero-oligomeric complexes with Co-SMAD (SMAD4), which are translocated to the nucleus where they regulate the expression of target genes. In general, both pathways are translocated into the nucleus to translate the signal [56]. In this recent study, the phosphorylation of SMAD2 was studied using Western blot analysis. Additionally, nuclear alteration was distinguished.

## 5. Conclusions

In conclusion, the encapsulation of TGFβ inhibitory peptides (P17 and P144) has shown potential therapeutic inhibition to basal phospho-SMAD as a result of TGF-β pathway regulation. It has proved clearly their ability to inhibit the autocrine TGFβ pathway and basal nuclear SMAD. However, SNU449 is one of the cell lines that is more responsive to those encapsulated by P17 and P144 is a mesenchymal, invasive HCC cell line that produces TGFβ autocrinally and is unresponsive to its suppressor effects. Additionally, rhodamine-labeled peptides enable researchers to follow nearly the localization of peptides inside cytoplasm after their cellular uptake. In the current studies and our previous studies indicated that encapsulation of TGFβ signaling pathway inhibitors may facilitate their delivery and overcome their cytotoxicity. Additionally, folic acid functionalized drug delivery systems enable them to reach and accumulate inside cancer cells [57,58,59,60,61,62].

## Figures and Tables

**Figure 1 pharmaceutics-12-00421-f001:**
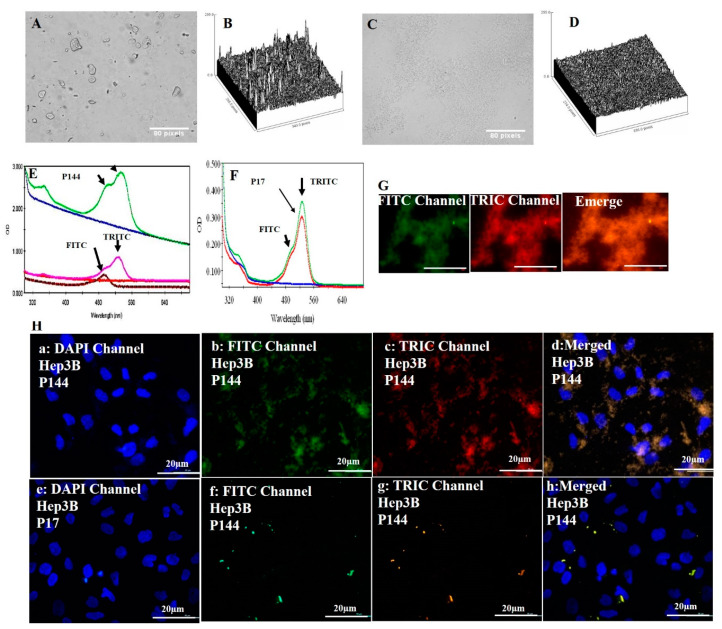
Optical microscopic image of P144 dissolution. (**A**) P144 dissolved by DMSO; (**B**) ImageJ analysis of P144 dissolved by DMSO; (**C**) P144 dissolved by sonication; (**D**) ImageJ analysis of sonicated P144; (**E**) rhodamine-labeled peptide P144; (**F**) rhodamine-labeled peptide P17; (**G**) fluorescence microscopy for images labeled P144. (**H**; **a**–**d**) Cellular uptake of sonicated P144; (E–H) Cellular uptake of suspended P17.

**Figure 2 pharmaceutics-12-00421-f002:**
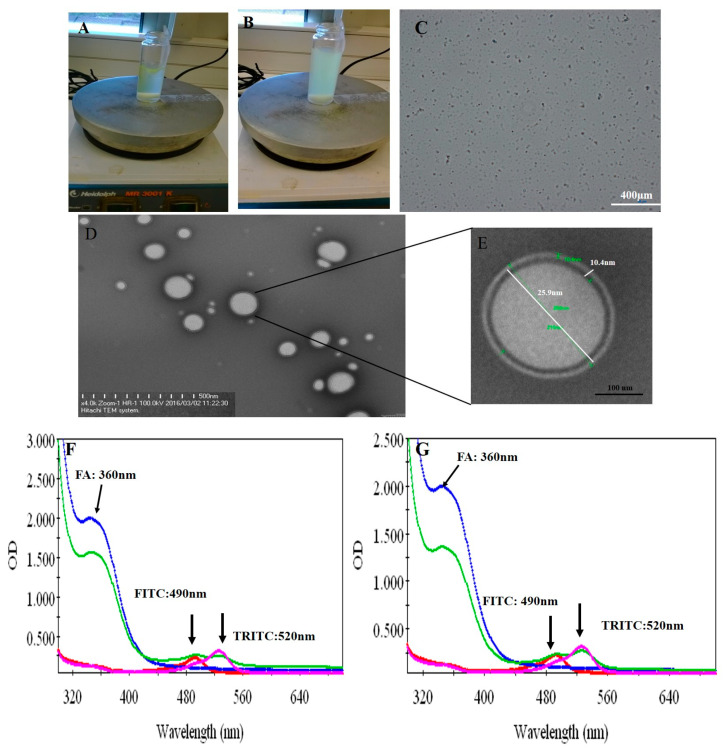
Fabrication of protein–peptide mucoadhesive complex. (**A**) Peptide conjugated protein; (**B**) turbid suspension after adding protamine. (**C**) Optical microscopic image reporting a homogenous distribution of particles. (**D**) Protein–peptide mucoadhesive complex TEM image. (**E**) TEM at high magnification. (**F**) UV–visible absorbance of P17. (**G**) UV–visible absorbance of P144.

**Figure 3 pharmaceutics-12-00421-f003:**
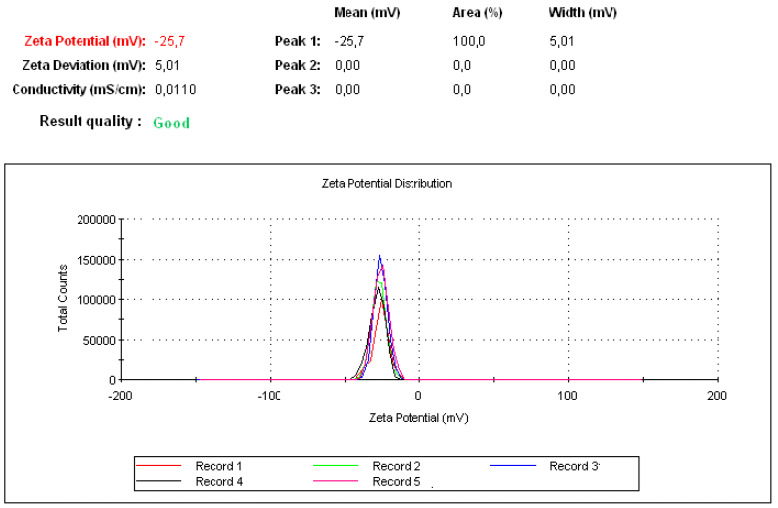
Zeta potential measurement after five successful running data. Each one was run five times with a delay of 5 s after each measurement.

**Figure 4 pharmaceutics-12-00421-f004:**
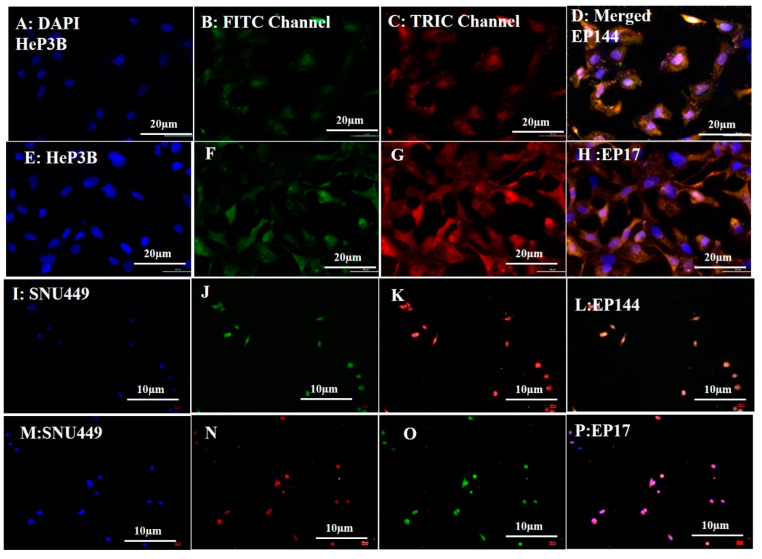
Fluorescence images of cellular uptake (**A**–**D**) Hep3B adsorbed EP144. (**E**–**H**) Hep3B adsorbed EP17. (**I**–**L**) SNU449-adsorbed EP144. (**M**–**P**) SNU449-adsorbed EP17.

**Figure 5 pharmaceutics-12-00421-f005:**
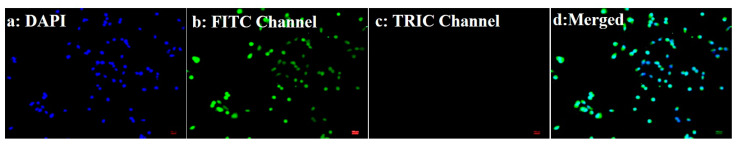
Fluorescence images of cellular uptake. SNU449-adsorbed free capsules.

**Figure 6 pharmaceutics-12-00421-f006:**
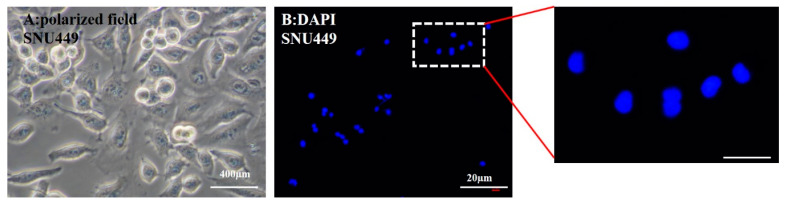
(**A**) Optical microscopy image of SNU449 after their exposure to 50 µg/mL free capsules for 24 h of incubation. (**B**) Fluorescence image of DAPI staining.

**Figure 7 pharmaceutics-12-00421-f007:**
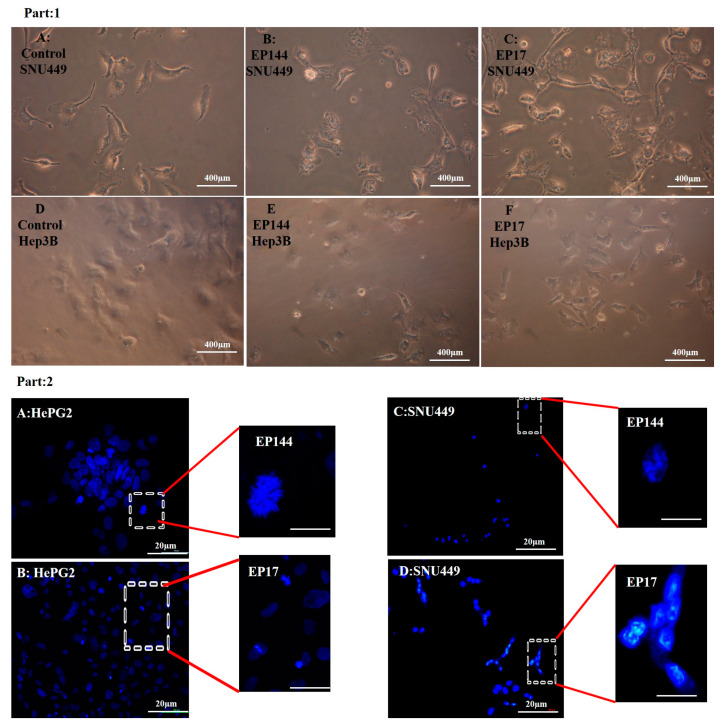
(**Part 1**) SNU449 and Hep3B cell lines suffered from phenotype changes. (**A**) control SNU449. (**B**) SNU449 exposed to 50 µg/mL EP144. (**C**)SNU449 exposed to 50 µg/mL EP17. (**D**) control Hep3B. (**E**) Hep3B exposed to 50 µg/mL EP144. (**F**) Hep3B exposed to 50 µg/mL EP17. (**Part 2**) Fluorescence images of morphology of DAPI staining (**A**) Nuclear degradation at Hep3B. (**B**) Nuclear condensation at Hep3B. (**C**) Nuclear fragmentation at SNU449. (**D**) Nuclear activation at SNU449.

**Figure 8 pharmaceutics-12-00421-f008:**
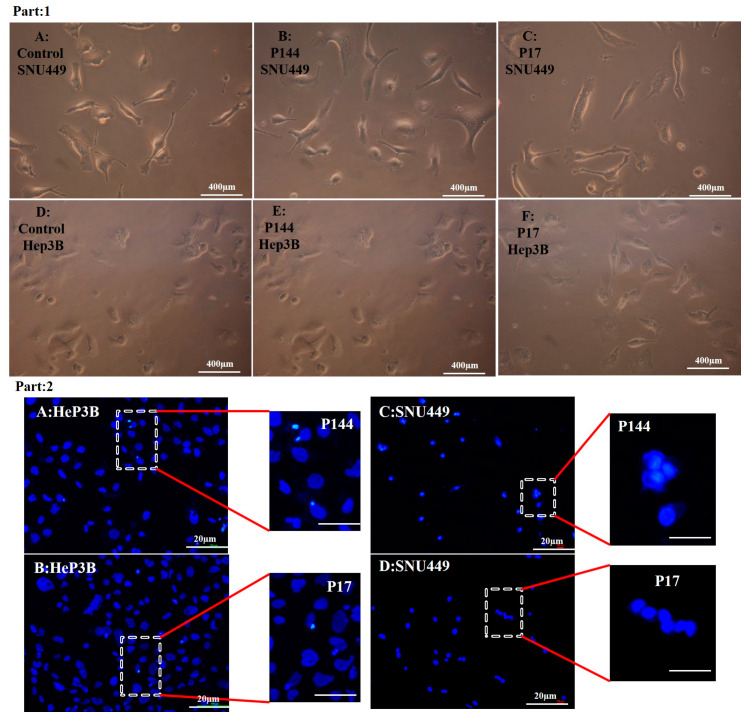
(**Part 1**) Optical microscopy images SNU449 and Hep3B cell lines suffered from phenotype changes. (**A**) control SNU449. (**B**) SNU449 exposed to 50 µg/mL P144. (**C**) SNU449 exposed to 50 µg/mL P17. (**D**) control Hep3B. (**E**) Hep3B exposed to 50 µg/mL P144. (**F**) Hep3B exposed to 50 µg/mL P17. (**Part 2**) Fluorescence images of the morphology of DAPI staining; (**A**) Slightly nuclear condensation at Hep3B. (**B**) Few nuclear condensations. (**C**) Slightly nuclear activation at SNU449. (**D**) No morphological alteration at SNU449.

**Figure 9 pharmaceutics-12-00421-f009:**
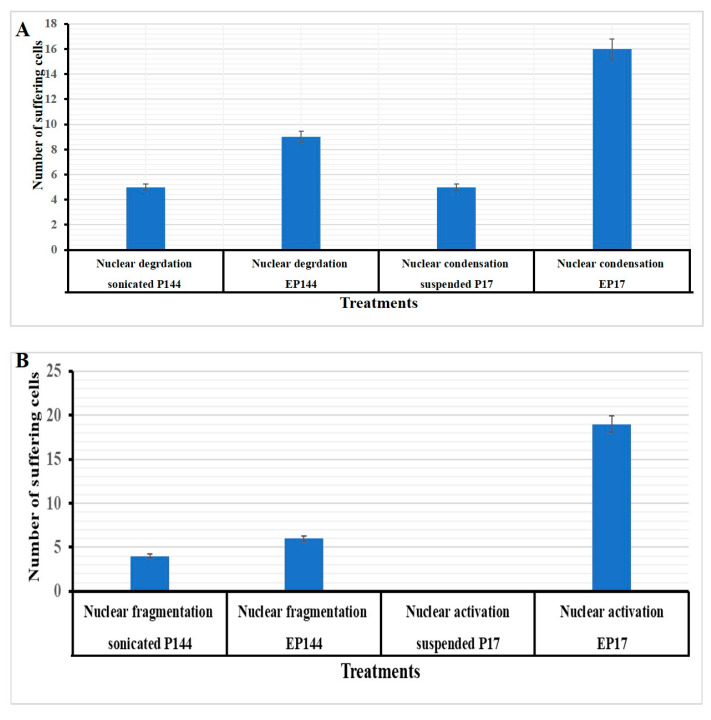
Quantification analysis of nuclear morphology after DAPI staining. (**A**) Hep3B cells exposed to encapsulated and non-encapsulated P17 and P144. (**B**) SNU449 cells exposed to encapsulated and non-encapsulated P17 and P144.

**Figure 10 pharmaceutics-12-00421-f010:**
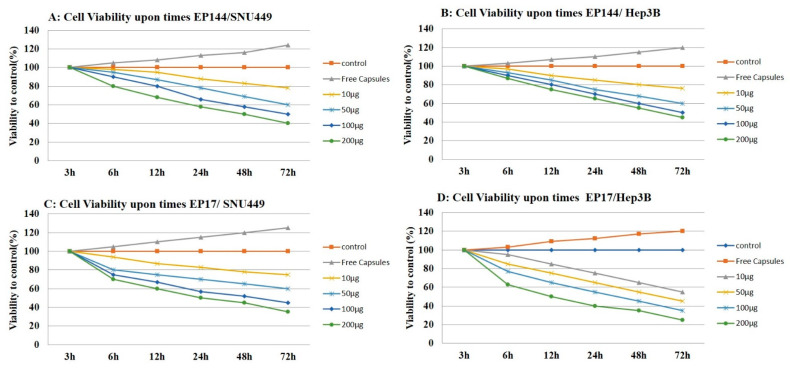
Cell proliferation assay. (**A**) Cell viability of SNU449 in a dose-dependent manner by using different concentrations of encapsulated P144. (**B**) Cell viability of Hep3B in a dose-dependent manner by using different concentrations of encapsulated P144. (**C**) Cell viability of SNU449 in a dose-dependent manner by using different concentrations of encapsulated P17. (**D**) Cell viability of Hep3B in a dose-dependent manner by using different concentrations of encapsulated P17.

**Figure 11 pharmaceutics-12-00421-f011:**
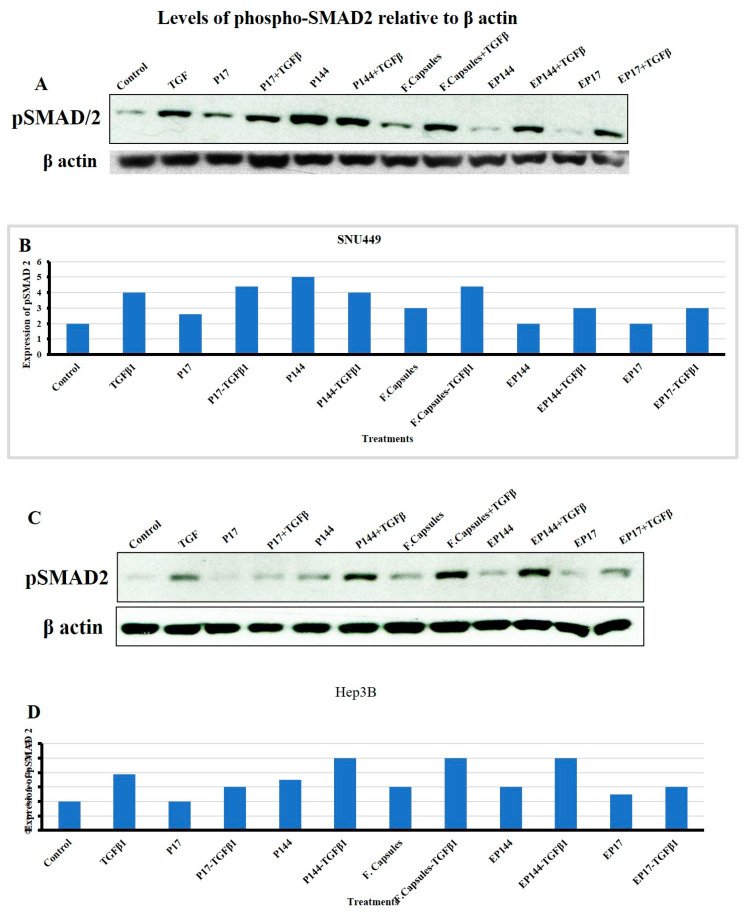
Evaluation of SMAD2 phosphorylation in SNU449 and Hep3B cells. (**A**,**C**) Western blot band of pSMAD2; beta-actin was analyzed as loading control. (**B**,**D**) Densitometric analysis of pSMAD2 levels after their exposure to several treatments for 6 h.

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
