# Peer review of "Encapsulating TGF-β1 Inhibitory Peptides P17 and P144 as a Promising Strategy to Facilitate Their Dissolution and to Improve Their Functionalization"

_pharmaceutics, 2020, doi:10.3390/pharmaceutics12050421_

Round 1

Reviewer 1 Report

In the manuscript “Encapsulating TGF-β1 inhibitory peptides P17 and P144 as a promising strategy to facilitate their dissolution and to improve their functionalization” authors describe application of peptide-protein complexes to increase peptide dissolution and cellular penetration and improve potential  therapeutic effect. It seems that the results confirm positive action of delivering peptides using BSA protein. However, authors should answer the following questions:

1. Authors write that P144 peptide is designed to block TGFβR III extracellular domains preventing cellular interaction between TGF ligand and its receptors. Why are they focusing on improving their internalization into cells? How the peptides block TGF signaling pathway inside the cell?

2. Why the authors claim that encapsulated peptides are in non-aggregated state after internalization into cells? The images from figure 1H (a-d) look very similar to the images from figure 4 A-D (Cellular uptake of P144).

3. How are the peptides released from the protein-peptide complex?

4. What is advantage of functionalization of the protein-peptide complexes with folic acid? The authors don’t show the control with cells with low level of folic acid expression.

5. There is a lack of comprehensive cytotoxicity studies of analyzed material. I would recommend to perform colorimetric assays to determine cytotoxic profile of the free capsules and protein-peptides in the dose-dependent manner to obtain optimal concentration range.

6. How the fluorescence signal is generated in FITC channel in peptides labeled with Rhodamine? Do the peptides possess intrinsic fluorescent properties in this range?

7. Why authors compare nuclear degradation (P144) vs nuclear condensation (P17) in Hep3B cells and nuclear fragmentation (P144) vs nuclear activation (P17) in SNU449 cells in Figure 8. These are different parameters. How the authors differentiate degradation from fragmentation. Some additional assays for nuclei evaluation should be performed.

8. There is a lack of proper physico-chemical characterization of protein-peptide complexes among others size distribution, zeta potential. I recommend to do DLS measurements or size evaluation from TEM images.

9. Authors write that they design nano-delivery strategies to provide protection for normal cells (line 20-21). However there is a lack of experiments performed on healthy cells.

10. Authors have to improve description for images. They are too general and sometimes is hard to deduce what they refer to.

11. There is a lack of the scale on images from fluorescence microscopy and TEM or it is very poor visible.

12. How was the images of granularity of peptides performed – figure 1B and 1D and figure 2D? How was it calculated. There is lack of description in Materials and Methods section.

13. The figures should be improved. For example western blot image is askew in figure 9A. Or in the UV-vis spectra in figure 1E and F there are more lines than description in the legends.

14. What does it mean that physical sonication was performed for 15 min under high amplitude (Power 5).

15. In Materials and Methods section the characterization part is very sparingly described. It is only a reference. The description should be more comprehensive.

16. In Materials and Methods section the Cellular uptake part is very poor described. The authors don’t write in which multi-well plates the experiments were performed and the number of cells per well in 6-well plate seems to be to low. They also don’t describe the experiment with DAPI.

17. There is a lot of typing errors throughout the manuscript.

Author Response

I would like to thank reviewer for his great efforts.

In the manuscript “Encapsulating TGF-β1 inhibitory peptides P17 and P144 as a promising strategy to facilitate their dissolution and to improve their functionalization” authors describe application of peptide-protein complexes to increase peptide dissolution and cellular penetration and improve potential therapeutic effect. It seems that the results confirm positive action of delivering peptides using BSA protein. However, authors should answer the following questions:

  1. Authors write that P144 peptide is designed to block TGFβR III extracellular domains preventing cellular interaction between TGF ligand and its receptors. Why are they focusing on improving their internalization into cells? How the peptides block TGF signalling pathway inside the cell?

I would like to thank the reviewer so much for this very great and interest question. SNU449 cells show autocrine stimulation of the TGFβ pathway and respond to TGF-β increasing Smad2/3 phosphorylation [31,40]. These cells are resistant to the suppressor effects of TGFβ [41] and respond to it undergoing EMT. Here we show that encapsulated peptides (P17 and P144) induced significant inhibition of basal pSMAD2 in SNU449 cells when compared to not-encapsulated peptides in absence of TGFβ stimulation. This result proved that peptides 17 and P144 have ability to inhibit TGFβ after their internalization by blocking autocrine stimulation of TGF β and to block basal nuclear SMAD.

  1. Why the authors claim that encapsulated peptides are in non-aggregated state after internalization into cells? The images from figure 1H (a-d) look very similar to the images from figure 4 A-D (Cellular uptake of P144).

I would like to thank reviewer.

This evidence was first detected by using fluorescence microscopy after their precipitation upon surface of slide (Figure 1,G). In the same way, small particles of both peptides were compacted with no equal distribution in perinuclear region after their internalization. To evaluate their aggregation, image J analysis was used and it show aggregation state in non-encapsulated peptide compared to encapsulated peptides (Supplementary, Figure S1)

  1. How are the peptides released from the protein-peptide complex?

I would like to thank reviewer.

Peptides attached protein can be released by enzymatic degradation in lysosomes

  1. What is advantage of functionalization of the protein-peptide complexes with folic acid? The authors don’t show the control with cells with low level of folic acid expression.

I would like to thank reviewer so much.  The internalization of nanoparticles marked by folic acid was demonstrated previously in our work against breast cancer and ovarian cancer cell lines. The result revealed that the successful accumulation of folic acid doped nanoparticles  inside cytoplasm of cancer cells depended mainly on  the overexpression of folate receptors [30]..

  1. There is a lack of comprehensive cytotoxicity studies of analyzed material. I would recommend to perform colorimetric assays to determine cytotoxic profile of the free capsules and protein-peptides in the dose-dependent manner to obtain optimal concentration range.

I would like to thank reviewer.

Crystal violet cell proliferation assay is a colorimetric method based on the use of crystal violet as a basic dye with avidity to nuclear structures. After binding and solubilization of the crystal violet, optical density measurements of extracted dye provide a measure of the relative number of viable cells. This test has the advantage that it is less time consuming, easier to perform and more objective [29].

Encapsulated peptides (P17 and P144) have antiproliferative influence on human liver cancer cell lines (SNU449 and Hep3B) as shown by proliferation assay. Both encapsulated peptides (P17 and P144) inhibited the proliferation of human liver cancer cell lines in vitro in a dose-dependent manner. This inhibition was substantial in both cell lines after treatment, with significant differences from 10 μg/mL to 200 μg/mL in treated cells (Figure 10). Similar to our findings, it is reported that P17 and P144 induced antiproliferation effect against Human lung cancer cell lines [44] and glioblastoma cell lines [45,46] respectively. On the other hand, Hep3B and SNU449 cell lines exhibited significant proliferation after their exposure to free capsules for 72h. 

  1. How the fluorescence signal is generated in FITC channel in peptides labelled with Rhodamine? Do the peptides possess intrinsic fluorescent properties in this range?

I would like to thank reviewer.

Such this emission is mostly related to presence of tryptophan, tyrosine and phenylalanine. While in the presence of rhodamine, the fluorescent signal is strengthened and intensity is increased. It is reported that tryptophan is high sensitive to hydrogen bonding and non-covalent interactions, displaying a red, green and blue shift [35]. Additionally, its indole group is considered the dominant source of UV absorbance at ∼280 nm and emission at ∼350 nm [36].

  1. Why authors compare nuclear degradation (P144) vs nuclear condensation (P17) in Hep3B cells and nuclear fragmentation (P144) vs nuclear activation (P17) in SNU449 cells in Figure 8. These are different parameters. How the authors differentiate degradation from fragmentation. Some additional assays for nuclei evaluation should be performed.

I would like to thank reviewer.

The results here just wanted calculate the nuclear alterations over all the samples (triplicates).This result proved  that the  alterations was distributed around  all of the samples, not only in the focusing place.

  1. There is a lack of proper physico-chemical characterization of protein-peptide complexes among others size distribution, zeta potential. I recommend to do DLS measurements or size evaluation from TEM images.

I would like to thank reviewer.

The physico-chemical characterization of protein peptide complexes was studied by using zeta potential measurement and TEM. Zeta potential of protein peptide mucoadhesive nanoparticles exhibited -25mV  (Figure 3) and their diameter was mostly ranged at nanoscale bar where they are captured by TEM Figure 2(D-E).  

  1. Authors write that they design nano-delivery strategies to provide protection for normal cells (line 20-21). However there is a lack of experiments performed on healthy cells.

I would like to thank reviewer so much. We are planning surely to investigate their ability to protect healthy cells by using animal model after treatment of Corona Virus infection.  

  1. Authors have to improve description for images. They are too general and sometimes is hard to deduce what they refer to.

I would like to thank reviewer. Despeciation of the images was improved

  1. There is a lack of the scale on images from fluorescence microscopy and TEM or it is very poor visible.

I would like to thank reviewer.  The scale bar was drawn upon all of images and the image of TEM is changed

  1. How was the images of granularity of peptides performed – figure 1B and 1D and figure 2D? How was it calculated. There is lack of description in Materials and Methods section.

I would like to thank reviewer.

2.2.5. Image j Analysis and calculation of the surface roughness

The measurements for the light microscopic images (400 µm) were carried out in pixels. Thus, setting up a scale bar for calibration for the calculation of the actual area was not required. Since, in the ‘Image option’ of ImageJ, under Adjust, ‘size” was selected. Then the pixel of width and length was adjusted and then it is followed by using the ‘Surface plot’ option in ‘Analyze’. This led to the area measurement of the complete image in pixels. The measured peaks upon material surface expresses the status of material that is either roughness or granules while no peaks were measured in case soft materials or no contain any granulated materials [26].

  1. The figures should be improved. For example western blot image is askew in figure 9A. Or in the UV-vis spectra in figure 1E and F there are more lines than description in the legends.

I would like to thank reviewer.  Western blot image is improved and the lines that were present at UV-Vis spectra, were removed

  1. What does it mean that physical sonication was performed for 15 min under high amplitude (Power 5).

 I would like to thank reviewer.

The sonication is operated to maximum amplitude (Power 5) to prevent formation of the air bubbles.

  1. In Materials and Methods section the characterization part is very sparingly described. It is only a reference. The description should be more comprehensive.

I would like to thank reviewer.  Text is changed as it follows:

Characterization

  • Transmission Electron Microscopy (TEM)

10 µL of nanoparticles suspension was deposited on the copper grid and air- dried before measurement. Copper grids sputtered with carbon films were used to support the sample. High-resolution TEM images of nanoparticles were analysed by a Hitachi HT 7700 operating at 100 kV, coupled with a GATAN camera ORIUS SC600 with a resolution of 7 Megapixel. The GATAN camera is controlled by Digital Micrograph [23].

2.2.3. UV-Vis spectroscopy

The absorbance of folic acid and Rhodamine was measured by using multiplate reader at option of UV Absorbance Spectrophotometer in multiplate wells. 500µl mL of fabricated nanoparticles was scanned at range 300–600 nm [24].

2.2.4. Zeta Potential

The electrophoretic mobility of samples was determined by photon correlation spectroscopy by using a Zeta Nano Sizer (Malvern Instruments, Malvern, UK). All measurements were performed at 25 °C. Five following measurements were taken for analysis, each one was run five times with a delay of 5 sec. after each measurement [25].

  1. In Materials and Methods section the Cellular uptake part is very poor described. The authors don’t write in which multi-well plates the experiments were performed and the number of cells per well in 6-well plate seems to be to low. They also don’t describe the experiment with DAPI.

I would like to thank reviewer.

2.3. Cell culture

Hep3B and SNU449 cells were obtained from the European Collection of Cell Cultures (ECACC). Cell lines were never used in the laboratory for longer than 4 months after receipt or resuscitation. All cells were maintained in DMEM media (Lonza, Basel, Switzerland) supplemented with 10% FBS (Sera Laboratories International Ltd, West Sussex, UK), Penicillin (100 U/ml), Streptomycin (100 μg/ml) and Amphotericin (2.5 μg/ml) and L-glutamine (2 mM). They were maintained in a humidified atmosphere at 37 °C and 5% CO2. Cells were observed under an Olympus IX-70 microscope.

2.3.1. Cellular uptake

Ten thousand of human liver cancer cell lines were seeded upon surface of sterilized cover slip,that was laid in the bottom of 6 multi-well micro plates. After 24 hours from their growth, encapsulated peptides and non-encapsulated (50 µg/ml) were added to each well and incubated in a humidified atmosphere of 37°C, 5% CO2.  SNU449 and Hep3B cell lines were fixed by 4% paraformaldehyde, then washed by PBS,7.2 (Phosphate Buffer Saline). Cells were then stained by DAPI (nuclear stain) for 30min and then washed twice by PBS7.2. Cellular uptake was analysedafter 24 h by red (TRITC), green (FITC) and blue (DAPI) channels of fluorescence microscopy. After that, images were acquired by fluorescence microscopy with a digital camera [27-28].

  1. There is a lot of typing errors throughout the manuscript.

I would like to thank reviewer.  The manuscript has been thoroughly revised

Reviewer 2 Report

The manuscript describes two peptides (P17 and P144)  designed  to block TGFbeta1 pathway.

But, unfortunately, the  main goal, for their therapeutic application, is strong limited by their cytotoxicity.

In addition, "the encapsulation of P17 and P144 might improve their therapeutic applications".

In my opinion the conclusions  must be improved and they, so maybe, supported by additional experimental evidences.

Minor point:

a) In Scheme 1 and 2  molecular strucures  could be better designed and ever more functional for Readers.

b) In References, you can find, many typing  errors (see numbers 11, 18, 22, 25, 26, 31, 38 and 44).

Author Response

I would like to thank reviewer  so much for his great efforts. 

Comments and Suggestions for Authors

The manuscript describes two peptides (P17 and P144) designed to block TGFbeta1 pathway.

But, unfortunately, the main goal, for their therapeutic application, is strong limited by their cytotoxicity.

I would like to thank reviewer so much. The cytotoxicity of encapsulated peptides (P17 and P144) was studied by anti-proliferative assay in presence of free capsules as described below.  

  • Antiproliferative effect of encapsulated peptides

Crystal violet cell proliferation assay is a colorimetric method based on the use of crystal violet as a basic dye with avidity to nuclear structures. After binding and solubilization of the crystal violet, optical density measurements of extracted dye provide a measure of the relative number of viable cells. This test has the advantage that it is less time consuming, easier to perform and more objective [29].

Encapsulated peptides (P17 and P144) have antiproliferative influence on human liver cancer cell lines (SNU449 and Hep3B) as shown by proliferation assay. Both encapsulated peptides (P17 and P144) inhibited the proliferation of human liver cancer cell lines in vitro in a dose-dependent manner. This inhibition was substantial in both cell lines after treatment, with significant differences from 10 μg/mL to 200 μg/mL in treated cells (Figure 10). Similar to our findings, it is reported that P17 and P144 induced antiproliferation effect against Human lung cancer cell lines [44] and glioblastoma cell lines [45,46] respectively. On the other hand, Hep3B and SNU449 cell lines exhibited significant proliferation after their exposure to free capsules for 72h. 

In addition, "the encapsulation of P17 and P144 might improve their therapeutic applications".

I would like to thank reviewer so much. Encapsulated peptides (P17 and P144) were accumulated and distributed  in perinuclear region of both Hep3B and SNU449 after their exposure compared to non-encapsulated peptides that were aggregated. Additionally, nuclear alterations was obtained in presence of encapsulated peptides whereas, less alteration was done in presence non-encapsulated peptides. The results confirm that the encapsulation improve therapeutic efficiency of both peptides. For instance, their encapsulation protect them from degradation, and control their delivery.  

In my opinion the conclusions must be improved and they, so maybe, supported by additional experimental evidences.

Accordingly it was improved as follows:

In conclusion, the encapsulation of TGFβ inhibitory peptides (P17 and P144) exhibits potential therapeutic inhibition to basal phospho-SMAD as a result of a TGF-β pathway regulation. Since, It is proven clearly their ability to inhibit autocrine TGFβ pathway and basal nuclear SMAD. Whereas, SNU449 as a one of the cell lines that is more responsive to these encapsulated P17 and P144 is a mesenchymal, invasive HCC cell line that produces TGFβ autocrinally and is unresponsive to its suppressor effects.  Additionally, Rhodamine labelled peptides enable researchers to follow nearly the localisation of peptides inside cytoplasm after their cellular uptake. In the current studies and our previous studies indicated that encapsulation of TGFβ signalling pathway inhibitors may facilitate their delivery and overcome their cytotoxicity. Additionally, folic acid functionalized drug delivery systems enable them to reach and accumulate inside cancer cells [57-60].

Minor point:

In Scheme 1 and 2 molecular strucures could be better designed and ever more functional for Readers.

I would like to thank reviewer so much. Both schemes have been transferred into supplementary

  1. b) In References, you can find, many typing errors (see numbers 11, 18, 22, 25, 26, 31, 38 and 44). 

    They were corrected as follows:

    11. Dotor, J.; Lopez-Vazquez, A.B.;Lasarte, J.J.;Sarobe, P.; Garcia-Granero, M.; Riezu-Boj, J.I.;Martinez,A.; ,Feijoo, E.;  Lopez-Sagaseta, J.; Hermida, J.;  Prieto, J.;  Borras-Cuesta, F. Identification of peptide inhibitors of transforming growth factor beta 1 using a phage‐displayed peptide library. Cytokine, 2007; 39: 106–15.

    18 Gil-Guerrero, L; Dotor, J.;  Huibregtse, I.L.;Casares, N.; López-Vázquez ,A.B.;  Rudilla, F.;  Riezu-Boj, J.I.;  López-Sagaseta, J.; Hermida, J.;  Van Deventer, S.; Bezunartea ,J.; Llopiz, D.; Sarobe ,P.; Prieto ,J.;  Borrás-Cuesta, F.;  Lasarte, J.J. In vitro and in vivo down-regulation of regulatory T cell activity with a peptide inhibitor of TGF-beta1. J Immunol ,2008; 181: 126– 35.

    22.Zhou, Q.Y.; Tu ,C.Y.; Shao, C.X.; Wang, W.K.; Zhu, J.D.;  Cai, Y.; Mao, J.Y.; Chen, W. GC7 blocks epithelial-mesenchymal transition and reverses hypoxia-induced chemotherapy resistance in hepatocellular carcinoma cells. Am. J. Transl. Res., 2017; 9: 2608–2617

    25.Parodi, A.; Miao, J.; Soond, S.M.; Rudzińska,M.R.; Zamyatnin, A.A. Albumin Nanovectors in Cancer Therapy and Imaging. Biomolecules ,2019, 9, 218.

    26.Scheicher, B.;  Lorenzer, C.; Gegenbauer, K.; Partlic, J.; Andreae, F.;  Kirsch, A.H.;  Rosenkranz, A.R.; Werzer, O.; Zimmer, A. Manufacturing of a Secretoneurin Drug Delivery System with Self-Assembled Protamine Nanoparticles by Titration. PLoS One,.2016 9; 11(11): e0164149.

    31.Chiba, K.;  Kawakami, K.; Tohyama, K.1998. Simultaneous evaluation of cell viability by neutral red, MTT and crystal violet staining assays of the same cells. Toxicol. In Vitro 12, 251e258.

    38.Beija, M.;  Afonso, A.M.; and  Martinho, M.G. Synthesis and applications of Rhodamine derivatives as fluorescent probes Chem. Soc. Rev. , 2009, 38(8); 2410–2433.

    44.Hanafy, N.A.; De Giorgi, M.L.; Nobile, C.; Cascione, M.; Rinaldi, R.;  Leporatti, S. CaCO3 rods as chitosan polygalacturonic acid carriers for brompyruvic acid delivery. Science of Advanced Materials,2016. 8(3); 514-523.

Reviewer 3 Report

The article is very interesting, But it needs some more data to substantiate the findings. 

Author Response

I would like to thank reviewer so much for his\her great interest. 

According to the abstract of pharmaceutics journal/ special issue entitled ”Nano-Micro encapsulation of drugs” and the aim of this special issue  that is announced “https://www.mdpi.com/journal/pharmaceutics/special_issues/encapsulation2020” to include paper describing the design, preparation, and characterization of nano and micro carrier based drug delivery  systems and their latest developments in this field.   We can confirm that our manuscript has covered most of the above issues. Hence, the fabrication of mucoadhesive protein peptide assembly was described step by step in result and discussion. Characterization was measured by using TEM, zeta potential and UV visible spectroscopy. Additionally, to investigate cellular uptake by using Hep3B and SNU449both peptides were labelled by Rhodamine and their internalization was measured in both case: in encapsulated and non-encapsulated state. Their drug efficiency was studied in dose dependent manner by using different concentrations (10, 50, 100, 200 µg/ml) for 3, 6, 12, 24, 48 and 72 h.)  Not only that, but also their inhibition for phosphor-SMAD was investigated as well. At moment, and because we have serious issues related to infection of Corona Virus, we are not allowed completely to go lab or even to follow any running experiments.

The comments of reviewer was reported step by step as follows:

  1. The authors nicely show the effect of pSMAD2, It is unclear about the biological significance of the pathway.

SMAD2 phophorylation is the first step in the TGF-beta signalling. Once the ligand binds to the receptor, SMAD2 is phosphorylated. The levels of phospho-Smad2 were studied after exposure of both Hep3B and SNU449 to encapsulated peptides in figure 7 part 1 and results indicate that both encapsulated peptides have much effective role on biological pathway compared to non -encapsulated peptides. While there is no phenotype inhibition was shown in case incubating SNU449 cell line with free capsules. Inhibition of SMAD2 phosphorylation reflects a complete inhibition in the TGF-beta pathway.

  1. It would be interesting to understand the migratory and invasive ability of the cells upon treatment with the respective peptides.

We would like to thank reviewer very much.  This point is very interesting. Paper here presented is the pharmacological and nanotechnological study for the encapsulation of the peptides and the demonstration of their efficiency. By sure, once the COVID-19 pandemic finishes, the suggested experiments will form part of our future studies.

  1. The authors can show the effect of the TGF-beta related genes to understand if they are regulated at the transcriptional level.

We would like to thank reviewer very much. In the same line of our response to the previous point, here we focused to study encapsulation of peptides inside mucoadhesive assembly (protein peptide formulation) and to study their characterization. Furthermore, nuclear alteration was shown by using DAPI staining and the alteration was described and the total changes were calculated. Nuclear alteration is not enough and it is not indicator to understand deeply the transcriptional level but may give insight to the influence of both encapsulated peptides on nuclear pathway, since all these effects depend on the transcriptional activity of the SMADs.

  1. A luciferase assay may be imperative to understand the effect.

Thanks again. Reporter assays to study SMAD-dependent transcriptional activity will be included in our next steps in the project, together with the analysis of gene expression previously mentioned. I would be very grateful if you accept this manuscript in the context of the design, preparation, and characterization of nano and micro carrier based drug delivery systems, as the issue asked for. Unfortunately, with the laboratories closed, we cannot do experimental work at that moment

  1. A schematic picture of where these peptides are binding is very important.

Thanks very much. We mentioned in the manuscript how these peptides were created and characterized. These previous works explain this point. We did not create the peptides. It was a company, DIGNA, in Spain. They characterized their mechanisms of action [13]..

  1. Are these peptides specific for TGFB1, did they test other ligands of the TGFB pathway.

We would like to thank reviewer very much. These two peptides (P17 and P144) were designed specifically to block TGF-beta ligands and to prevent the connection between TGF-beta ligands and their receptors. Peptide P144; TSLDASIIWAMMQN, is a very hydrophobic peptide obtained from the membrane-proximal ligand-binding domain of b-glycan [9]. This peptide is designed to block TGFβR III extracellular domains preventing cellular interaction between TGF ligand and its receptors [10].  Another soluble peptide is called P17, (KRIWFIPRSSWYERA) [11]. It was produced from a phage library [12]. P17 can block TGF-β1, TGFβ2, and TGFβ3 with relative affinity binding reached 100%, 80% and 30% respectively [13].

The transforming growth factor β (TGF-β) signalling pathway consists of extracellular ligands (including TGF-β-like group and BMP-like group). TGF-β-like group generally phosphorylate SMAD2 and SMAD3, whereas BMP-like group generally induce phosphorylation of SMAD1, SMAD5 and SAMD8[55].. The activated R-SMADs form hetero-oligomeric complexes with Co-SMAD (SMAD4), which are translocated to the nucleus where they regulate the expression of target genes [56].. In general, the both pathways are translocated into nucleus to translate the signal. In this recent study, the phosphorylation of SMAD2 was studied  as by using western blot analysis.  Additionally the nuclear alteration was distinguished.

  1. Can the authors check the effect on non SMAD signaling like the p38 pathway.

We would like to thank reviewer very much. We focus the discussion mostly on SMAD  pathways activated by TGF-β at moment because and according to our knowledge, there is limitation in literature review concerning encapsulation of TGF-beta inhibitory peptide. This question could open our understanding in future to study  other non smad pathway such as extracellular signal-regulated kinases (Erks), c-Jun amino terminal kinase (JNK), p38 MAPK, as well as the IκB kinase (IKK), phosphatidylinositol-3 kinase (PI3K) and Akt, and Rho family GTPases. In HCC cells it is also very relevant the interaction between the TGF-beta pathway and the EGFR pathway. This is another pathway that could be explored.

Round 2

Reviewer 1 Report

Authors have answered all questiones and accepted most suggestions. I have only one additional question: what does it mean that the zeta potential value is good (line 222)? In which context it is good?

Author Response

I would like to thank reviewer so much. This indicates good physical stability of nanosuspensions due to electrostatic repulsion of individual particles.

Reviewer 2 Report

Authors have accepted all suggestions and they have also edited the manuscript accordingly .

Author Response

We would like to thank reviewer  so much for his/her great efforts during revision our manuscript.

Reviewer 3 Report

 The article is interesting and sheds some light on a unique peptide for inhibiting the tgfb pathway. 

Author Response

We would like to thank reviewer so much for his/her great efforts during revision our manuscript
